# Molecular polarizability anisotropy of liquid water revealed by terahertz-induced transient orientation

Peter Zalden[1,2], Liwei Song[3,4], Xiaojun Wu[3,5], Haoyu Huang[3], Frederike Ahr[3], Oliver D. Mücke [1,3], Joscha Reichert[1,6], Michael Thorwart[1,6], Pankaj Kr. Mishra[1,3], Ralph Welsch [3], Robin Santra [1,3,7], Franz X. Kärtner[1,3] & Christian Bressler[1,2]

Reaction pathways of biochemical processes are influenced by the dissipative electrostatic interaction of the reagents with solvent water molecules. The simulation of these interactions requires a parametrization of the permanent and induced dipole moments. However, the underlying molecular polarizability of water and its dependence on ions are partially unknown. Here, we apply intense terahertz pulses to liquid water, whose oscillations match the timescale of orientational relaxation. Using a combination of terahertz pump / optical probe experiments, molecular dynamics simulations, and a Langevin dynamics model, we demonstrate a transient orientation of their dipole moments, not possible by optical excitation. The resulting birefringence reveals that the polarizability of water is lower along its dipole moment than the average value perpendicular to it. This anisotropy, also observed in heavy water and alcohols, increases with the concentration of sodium iodide dissolved in water. Our results enable a more accurate parametrization and a benchmarking of existing and future water models.

[1] Centre for Ultrafast Imaging CUI, University of Hamburg, 22761 Hamburg, Germany. [2] European XFEL, Holzkoppel 4, 22869 Schenefeld, Germany. [3] Center for Free-Electron Laser Science CFEL, Deutsches Elektronen-Synchrotron, 22607 Hamburg, Germany. [4] State Key Laboratory of High Field Laser Physics, Shanghai Institute of Optics and Fine Mechanics, Chinese Academy of Sciences, 201800 Shanghai, China. [5] School of Electronic and Information Engineering, Beihang University, 100191 Beijing, China. [6] I. Institut für Theoretische Physik, University of Hamburg, Jungiusstr. 9, 20355 Hamburg, Germany. [7] Department of Physics, University of Hamburg, Jungiusstr. 9, 20355 Hamburg, Germany. These authors contributed equally: Peter Zalden, Liwei Song. Correspondence and requests for materials should be addressed to P.Z. (email: peter.zalden@xfel.eu)

D espite their omnipresence and their relevance in biochemistry, fundamental properties of water molecules are not yet known[1]. One such example is the polarizability tensor $\alpha_{ij}$, which, together with the permanent dipole moment $\mu_i$, governs the interaction of water molecules with an electric field $E_i$ via their energy

$$W = -\sum \mu_i E_i - \frac{1}{2} \sum \sum \alpha_{ij} E_i E_j - \ldots \qquad (1)$$

This fundamental interaction determines the outcome of chemical and biochemical reactions, e.g., in proteins[2,3], whose folding reaction was calculated using classical force field models[4]. While these models can reproduce the molecular structure obtained from ab-initio simulations, the polarization energy can only be obtained when the anisotropic polarizability due to the oxygen lone pairs is included[5]. Furthermore, comparative studies of polarizable and non-polarizable water models have shown that the latter are unable to reproduce fundamental properties of water, such as the heat capacity, vapor pressure, and dielectric constant[6,7]. Hence, an accurate water model requires reliable data for the molecular polarizability tensor in the liquid state. In this work, we use the Kerr effect to determine $\Delta\alpha = \alpha_\parallel - \alpha_\perp$, i.e., the difference of the polarizability along the permanent dipole moment, $\alpha_\parallel$, and the average value perpendicular to it, $\alpha_\perp$. The yet unknown sign of $\Delta\alpha$ determines whether the polarizability (and refractive index) of water increases or decreases upon application of an electric field that orients the dipole moments.

For water molecules in the gas phase, two possible sets of values with opposite signs were previously derived from the Raman spectrum of rotational modes[8]. By comparison with computer simulations, it was found that $\Delta\alpha < 0$ and not $\Delta\alpha > 0$ is the more likely result. Based on present-day literature, it is unclear whether $\Delta\alpha > 0$[9] or $\Delta\alpha < 0$[10] is the correct result, and an abundance of references exists for either case[11]. This is mostly due to the small value of the anisotropy $\Delta\alpha \ll \overline{\alpha}$ as compared to the isotropically averaged polarizability of water molecules. Nevertheless, these values, obtained on water monomers, are commonly used in liquid water models[12]. For the liquid state, computer simulations commonly predict $\Delta\alpha > 0$[11,13]. In contrast, LeFèvre et al. proposed already in 1960 that the molecular Kerr coefficient of water could be negative, implying $\Delta\alpha < 0$[14]. Therefore, the sign of $\Delta\alpha$ for liquid water is presently unclear. It is worth mentioning that studies using DC electric fields commonly find a reduction of the refractive index in the direction of the applied field due to dielectric saturation[15–17].

The optical Kerr effect (OKE) is a well-established technique to measure the modulus of the anisotropy of the polarizability tensor in molecular liquids[18,19]. Therefore, the sign of $\Delta\alpha$ cannot be determined using the OKE. This is because the orientation of permanent dipole moments cannot follow the direction of the rapidly oscillating optical field[19]. This implies that the first term in Eq. 1 is negligible and the second term dominates, inducing an alignment of the molecules' axis of highest polarizability parallel to the electric field. Therefore, the resulting Kerr effect must be positive—at least when the optical probe frequency matches the frequency of the driving field or more generally, when it is separated from resonances in the dielectric function. This condition is commonly fulfilled in OKE experiments. In case of water it is well known that the axis of highest polarizability is in the direction spanned by the two hydrogen atoms, i.e., perpendicular to the permanent dipole moment[8]. Hence, the OKE cannot be employed to determine $\Delta\alpha$. Indeed, the polarizability anisotropies were not reported in previous studies of the OKE in water[20–22]. In the terahertz (THz) regime, however, the coupling is dominated by the interaction with the permanent dipole moment $\mu$ (as

evidenced later by our molecular dynamics (MD) simulations and the experimental results) and therefore orients the permanent dipole moments in the direction of the THz field[23]. Thus, the so-called THz-induced Kerr effect (TKE) should enable the first experimental determination of $\Delta\alpha$ of water in the liquid state. The dielectric function of water in the frequency range below 2 THz is known to be dominated by relaxation mechanism, and is often decomposed into three distinct mechanisms RI−RIII[24–28] with Debye relaxation times $\tau_D$ of 9.01, 1.03, and 0.085 ps, respectively at 23 °C[27]. The microscopic nature of these relaxation mechanisms is not fully resolved. Alternative models for the water dynamics in this frequency regime are based on angular jumps[29] or rotational double-well potentials[28], but there is consensus that the dynamics are caused by angular relaxations of water molecules.

Single-cycle THz Kerr spectroscopy was recently pioneered by Hoffmann et al., who found that the birefringence in liquids such as carbon disulfide ($CS_2$) prevails even after the driving electric field is no longer present[23,30–34] and relaxes on a longer timescale of 1.7 ps, which matches the timescale for the loss of collective orientation, 1.64 ps[35]. Recent TKE experiments on water vapor show that $\Delta\alpha > 0$ due to the positive sign of the Kerr effect[23]. The application of this technique to liquid water and alcohols is challenging due to their orders of magnitude higher absorption and lower Kerr coefficients[36]. Therefore, these experiments require THz sources of highest field strengths and lowest pulse-to-pulse fluctuations at the same time. Furthermore, the group velocity mismatch of the THz and optical pulses must be considered during the data analysis.

Here, we induce the Kerr effect in liquids with single-cycle electromagnetic pulses at 0.25 THz center frequency. The resulting birefringence $\Delta n = n_\parallel - n_\perp$ has a molecular contribution, which we identify by comparison with the electric field waveform $E(t)$ obtained by electro-optic (EO) sampling. In case of liquid water and alcohols, we find this contribution to be negative. Molecular dynamics simulations and a solution of the Langevin equation show that the THz electric field induces an orientation of the permanent dipole moments of water along the field. This implies that the polarizability of water molecules is smaller parallel to the dipole moment than the average value perpendicular to it. Dissolving sodium iodide in water enhances the amplitude of the negative Kerr signal and therefore enhances the anisotropy of the polarizability tensor of water.

## Results

**The Kerr effect in polar liquids**. In our experiment, the Kerr effect is induced by single-cycle electromagnetic THz pulses $E(t)$ with electric field strength up to 510 kV/cm (see "Methods" section for details and Supplementary Note 1). The resulting birefringence is probed by the phase shift $\Delta\phi(t)$ of a co-propagating optical pulse of 150 fs duration and wavelength $\lambda = 800$ nm with linear polarization tilted 45° against the polarization of the THz field. The measured phase shift $\Delta\phi(t)$ is caused by the birefringence $\Delta n(z, t)$ at position $z$ along the direction of propagation, via

$$\Delta\phi(t) = \frac{2\pi}{\lambda} \times \int_0^L \Delta n(z, t) \mathrm{d}z, \qquad (2)$$

where $L$ denotes the thickness of the sample. Furthermore, $\Delta n$ can be decomposed into two contributions—a temperature-independent electronic effect,

$$\Delta n_e(z, t) = \lambda B_e E^2(z, t), \qquad (3)$$

which follows the square of the instantaneous THz electric field,

and a temperature-dependent molecular mechanism. We model the latter from a Langevin dynamics model[37], derived in Supplementary Note 4 on the basis of Brownian motion of a molecule in the overdamped and dilute limit (Supplementary equation 19) under the assumption of an isotropic rotational diffusion tensor,

$$\Delta n_{\mathrm{m}}(z,t) = \frac{2\pi B_{\mathrm{m}}^{(1)}}{\tau_2} \beta \int_{-\infty}^{t} \mathrm{d}t'\, E^2(z,t') \times \exp\left(-\frac{t-t'}{\tau_2}\right)$$
$$+ \frac{2\pi B_{\mathrm{m}}^{(2)}}{\tau_1 \tau_2} \beta^2 \int_{-\infty}^{t} \mathrm{d}t''\, E(z,t'') \times \exp\left(-\frac{t-t''}{\tau_2}\right)$$
$$\times \int_{-\infty}^{t''} \mathrm{d}t'\, E(z,t') \times \exp\left(-\frac{t'-t''}{\tau_1}\right), \quad (4)$$

where the birefringence at time $t$ is influenced by the THz electric field at all earlier times $t', t'' < t$ and $\beta = (k_{\mathrm{B}}T)^{-1}$. The relaxation times are related to the Debye relaxation time via $\tau_1 = \tau_{\mathrm{D}}$ and $\tau_2 = \tau_{\mathrm{D}}/3$. Based on the underlying assumptions, we do not expect that Eq. 4 will necessarily reproduce the exact dynamics of the TKE, especially for those liquids with pronounced anisotropy or in case correlation effects become relevant. Nevertheless, Eq. 4 is a reasonable approximation to the dominating physics.

Specifically, the two terms scaled by $B_{\mathrm{m}}^{(1)}$ and $B_{\mathrm{m}}^{(2)}$ correspond to polarization-induced alignment and dipole moment-induced orientation, respectively, as discussed in the following. The coefficient $B_{\mathrm{m}}^{(1)} = c_1\left(\Delta\alpha \times \Delta\varepsilon + \frac{3}{4} \times \Delta\alpha^+ \times \Delta\varepsilon^+\right)$, with $c_1 > 0$. $\alpha_{ij}$ is the optical polarizability tensor at the probe frequency and $\varepsilon_{ij}$ is the dielectric tensor at the THz pump frequency. $\Delta\alpha = \alpha_{zz} - 0.5 \times (\alpha_{xx} + \alpha_{yy})$ and $\Delta\alpha^+ = \alpha_{xx} - \alpha_{yy}$, where $z$ is defined as the axis of the permanent dipole moment; or in the absence of a dipole moment as the axis of rotational symmetry. Since $\Delta\varepsilon$ is defined analogously and the overall shape of both tensors is expected not to differ between the THz and optical regime, $B_{\mathrm{m}}^{(1)}$ effectively scales with $\Delta\alpha^2$ and is therefore positive. This term describes the alignment of molecules with their axis of highest polarizability parallel to the THz electric field. The coefficient $B_{\mathrm{m}}^{(2)} = c_2\Delta\alpha\mu^2$, with $c_2 > 0$, and $\mu$ is the permanent dipole moment of the molecule under consideration. This term describes an orientation of the molecule with its permanent dipole moment in the direction of the electric field vector. It enables the determination of the sign of $\Delta\alpha$ based on the sign of $\Delta n_{\mathrm{m}}$ alone. This is possible since the temporal average of the term scaling with $B_{\mathrm{m}}^{(2)}$ in Eq. 4 always has the sign of $B_{\mathrm{m}}^{(2)}$ itself ($\int \mathrm{d}t E(t) = 0$ for any THz pulse propagating in free space) and in case of all polar liquids discussed here, it dominates over the one with $B_{\mathrm{m}}^{(1)}$ (refs. [23,33], as well as Supplementary Note 4). Hence, for polar molecules $\Delta n_{\mathrm{m}} \propto \Delta\alpha$. The electronic contribution $\Delta n_e$ at THz frequencies instantaneously follows $E^2(z,t)$ and originates from a field-induced modification of the molecular polarizability, i.e., from hyperpolarizabilities, which are known to be small in case of water[10].

**Modeling experimental data.** We emphasize that the decomposition into an electronic $B_e$ and a molecular contribution $B_{\mathrm{m}}$ to the Kerr effect dynamics is possible when the electric field waveform $E(z,t)$ is known. Alternatively, the temperature dependence of the molecular Kerr effect can be used to decouple it from the temperature-independent electronic mechanism[19]. The electric field waveform can be measured due to the stable carrier-envelope phase (CEP) of the THz pulses, and the incident THz electric field $E(z=0, t)$ is obtained directly and in absolute units via EO sampling. In a second step, we calculate $E(z,t)$, the propagation of the THz pulse through the cuvette and the liquid. In this calculation, we consider

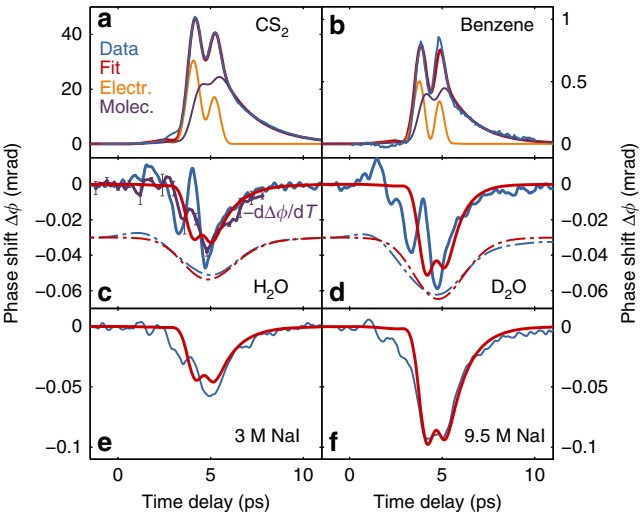

**Fig. 1** The THz-induced Kerr effect in water and reference liquids. The refinement (red curve) is the sum of two contributions: An instantaneous electronic birefringence (orange curve) and a delayed, molecular birefringence (purple curve). While in the non-polar molecules carbon disulfide ($CS_2$) and benzene (**a**, **b**), a positive molecular alignment effect is observed together with a positive electronic response, the polar water molecules in regular ($H_2O$; **c**) and heavy ($D_2O$; **d**) water as well as aqueous solutions of sodium iodide (NaI; **e**, **f**) reveal a negative molecular orientation effect. The electronic Kerr effect background of the cuvette was subtracted. We also isolate the molecular orientation mechanism of water based on its temperature dependence $-\mathrm{d}\Delta\phi/\mathrm{d}T$ (purple curve) in a background-free measurement. Error bars correspond to the standard deviation. In **c**, **d**, additional dotted curves, offset by $-0.03$ mrad, correspond to a convolution with a Gaussian $\sigma = 1.4$ ps. Error bars represent the standard deviation

the full dielectric function of each material when solving the Fresnel equations for the interfaces and when propagating the pulse in the frequency domain—described further in Supplementary Note 1. To enable this procedure, we have measured the dielectric functions of all liquids under investigation and depict the results in Supplementary Figs. 6 and 7. Since we do not expect Eq. 4 to render the exact dynamics, we restrict ourselves to a refinement of $B_{\mathrm{m}}^{(2)}$ in the case of polar molecules. In this way, we consider the inherent smallness of $\Delta\alpha$ against the expected dipole moments for the substances under investigation: as $B_{\mathrm{m}}^{(1)}$ is expected to scale with an additional factor $\Delta\alpha$, we expect its contribution to be comparably small in contrast to $B_{\mathrm{m}}^{(2)}$. Indeed, for the case of liquid water, a comparison of prefactors suggests a factor of $10^{-4}$ difference in signal strength (see Supplementary Note 4).

**THz Kerr effect in reference liquids.** Measurements on carbon disulfide ($CS_2$) and benzene ($C_6H_6$) (blue curves in Fig. 1a, b) show that the model accurately reproduces the TKE in these liquids. The red curve in Fig. 1 represents the overall model $\Delta\phi(t)$ in Eqs. 2–4, while the orange and purple curves correspond to the electronic and molecular contributions, respectively. The resulting three parameters $B_e$, $B_{\mathrm{m}}^{(1)}$, and $\tau$, which are used for the refinement, are summarized in Table 1. The positive sign of the molecular contribution $B_{\mathrm{m}}$ in both of these liquids confirms our earlier discussion for molecules without permanent dipole moment, since $\Delta\alpha > 0$ for $CS_2$, while $\Delta\alpha < 0$ for $C_6H_6$[38].

**THz Kerr effect in water.** The measured Kerr effect for liquid water is shown in Fig. 1c, blue curve, and was verified to scale

**Table 1 Summary of literature and experimental parameters relevant to this study**

| | $n$ ($\lambda = 800$ nm) | $n$ ($\lambda = 800$ μm) (0.37 THz) | $\alpha$ ($\lambda = 800$ nm) in cm$^{-1}$ | $B_{opt}$ in $10^{-14}$ m/V$^2$ ($\lambda_{pump}/\lambda_{probe}$) | $B_{stat}$ in $10^{-14}$ m/V$^2$ | $B_e$ in $10^{-14}$ m/V$^2$ | $B_m^{(1)}$ in $10^{-14}$ m/V$^2$ | $B_m^{(2)}$ in $10^{-14}$ m/V$^2$ | $B_m/B_e$ | $\tau_2$ in ps | $B_e/B_{opt}$ |
|---|---|---|---|---|---|---|---|---|---|---|---|
| CS$_2$ | 1.6058[54] | 1.70 | <0.5 | 4.1 (694/488)[20] | 3.24[55] | 0.28 | 0.22 | – | 0.77 | 1.84 | 0.07 |
| 2-Propanol | 1.379[56] | 1.54 | 22 | 0.052 (1064/442)[20] | 3.1[57] | 0.0097 | – | −0.0033 | −0.34 | 1.52 | 0.19 |
| Ethanol | 1.3573[54] | 1.60 | 22 | 0.051 (1064/442)[20] | 0.44[57] | 0.0093 | – | −0.0067 | −0.72 | 0.96 | 0.18 |
| Methanol | 1.323[56] | 1.80 | 64 | 0.034 (1064/442)[20] | 2.17[57] | 0.016 | – | −0.018 | −1.14 | 1.46 | 0.47 |
| H$_2$O | 1.3282[54] | 2.48 | 143 | 0.035 (694/488)[20] | 2.92[58] | <0.003 | – | −0.025 | < −8.3 | 1.1 | <0.1 |
| H$_2$O + 1 M NaI | 1.347[59] | 2.51 | 160 | | | <0.003 | – | −0.027 | < −9.1 | 1.1 | |
| H$_2$O + 3 M NaI | 1.388[59] | 2.63 | 166 | | | <0.003 | – | −0.043 | < −14.3 | 1.1 | |
| H$_2$O + 5 M NaI | 1.420[59] | 2.76 | 165 | | | <0.003 | – | −0.071 | < −23.8 | 1.0 | |
| H$_2$O + 9.5 M | 1.475[59] | 2.91 | 157 | | | <0.003 | – | −0.102 | < −33.9 | 1.0 | |
| D$_2$O | 1.324[54] | 2.34 | 130 | 0.029 (1064/442)[20] | | <0.003 | – | −0.021 | < −6.9 | 1.36 | <0.1 |
| Fused silica | 1.453[60] | 1.95 | 1.5 | 0.018 (1064/1064)[61] | | 0.0033 | 0.0004 | – | 0.13 | 2.38 | 0.18 |
| Benzene | 1.489[62] | 1.51 (1.51[63]) | ~1 (1.42[63]) | 0.70 (694/488)[20] | 0.39[55] | 0.035 | 0.029 | – | 0.84 | 1.55 | 0.05 |

$B_{opt}$ is the Kerr coefficient derived from OKE experiments

with the square of the THz pump field (see Supplementary Note 2). Figure 1c also shows a refinement of the model in Eqs. 2–4, red curve. While the changes in refractive index during the application of the field are not well described, the relaxation behavior is accurately reproduced with $\tau_2 = 1.1$ ps. During the refinement process, $\tau_1 < \tau_2/10$ is found, because otherwise the molecular Kerr effect shows bipolar oscillations, not observed experimentally. In this case, the first and second terms in Eq. 4 obtain approximately the same functional form. To further support the decoupling of the Kerr effect of water into an electronic and a negative molecular contribution, the measurement was repeated at various temperatures $T$ between 23 and 68 °C (see Supplementary Note 6). The resulting temperature-dependent signal, $-d\Delta\phi/dT \propto \Delta\phi_m$ (purple curve) scales with the molecular contribution to the Kerr effect alone. This implies that any influence from the electronic effects inside the cuvette and from water are not relevant. Indeed, the Kerr signal of an empty cuvette is confirmed to be temperature-independent. On the other hand, the resulting temperature dependence for a cuvette filled with water is in good agreement with the earlier decoupling by refining Eqs. 2–4 based on the known electric field $E(z, t)$. We conclude that the residual signal, not observed in the temperature dependence, originates from electronic effects in water and in the cuvette. Therefore, the average TKE in water is found to be negative, as can also be seen from the dashed curves, which are obtained after convolution with a Gaussian of width $\sigma = 1.4$ ps and correspond to the results expected when performing the experiment with longer probe pulses, not resolving the fast dynamics. This observation is consistent with recent experimental TKE data after excitation at around 2 THz[36]. According to Eqs. 2–4, the negative sign of the Kerr effect can only be obtained by a negative sign of $\Delta\alpha$, which then causes the molecular orientation term $B_m^{(2)} \propto \mu^2 \Delta\alpha$ to become negative. The molecular alignment term $B_m^{(1)} \propto \Delta\alpha^2$, on the other hand, can only cause a positive birefringence. In this model, the THz-induced perturbation is assumed to be dominated by an orientation of the dipole moments of individual water molecules along the electric field of the THz pulse (see also the solution of the full Langevin model in Supplementary Note 4). To clarify the role of correlated rotations of multiple water molecules[39,40], we have performed molecular dynamics (MD) simulations on water based on the rigid TIP4P/2005 force field[41], explicitly including the time-dependent electric field of the THz pulse (see Supplementary Note 5). The simulations confirm that the THz pulses induce an orientation of the water molecules along the field: Let $\theta$ be the angle between the

polarization axis of the THz electric field and the permanent dipole moment of one water molecule. $\Delta n$ scales, among other factors independent of $\theta$, with $\langle\cos^2(\theta)\rangle - 1/3$, where $\langle\ldots\rangle$ denotes the ensemble average[42]. $\langle\cos^2(\theta)\rangle - \frac{1}{3} > 0$ is caused both by alignment and orientation, but $\cos\theta \neq 0$ is observed only when the water molecules are oriented[32]. The results of our MD simulations are summarized in Fig. 2, explicitly showing that not only $\langle\cos^2(\theta)\rangle - \frac{1}{3}$ but also $\langle\cos(\theta)\rangle$ is non-zero and therefore, the water molecules are indeed oriented with their dipole moments along the THz electric field. Consistent with the analytic model, the maximum orientation of the molecules is achieved with a time lag of a few hundred fs in comparison to the electric field profile of the pulse. A similar effect can also be observed for $\langle\cos^2(\theta)\rangle$, which scales with the experimental observable $\Delta n \propto \Delta\alpha \times (\langle\cos^2(\theta)\rangle - 1/3)$, so that a negative $\Delta\alpha$ explains the negative sign of the TKE. The employed THz pulse in the frequency range of 0.3–3 THz couples to the collective modes of water connected by hydrogen bonds[39,40] and, as the field amplitude increases, the hydrogen bond network is weakened enough to allow for orientation. To further support the existence of an underlying orientation mechanism of the permanent dipole moments, we have evaluated the Langevin model based on the fully anisotropic rotational diffusion tensor reported for liquid water in literature. The resulting $\langle\cos\theta\rangle$ in Supplementary Fig. 8 is in good agreement with the result of the MD simulation, indicating that correlation effects included in the MD simulation, but not included in the Langevin model, do not dominate the THz-induced Kerr effect of water.

The TKE in water, observed experimentally, is best described by the choice of parameters $B_m^{(2)} = -0.025 \times 10^{-14}$ mV$^{-2}$, $|B_e| < 0.003 \times 10^{-14}$ mV$^{-2}$ and $\tau_2 = 1.1$ ps, as also shown in Table 1. The upper bound for $|B_e|$ corresponds to the value at which the residual (root mean square) discrepancy between data and model doubles. We can conclude that the molecular coefficient $B_m^{(2)} = -0.025 \times 10^{-14}$ mV$^{-2}$ is of larger magnitude than hyperpolarizability effects described by $B_e$. The time constant corresponds to a Debye relaxation time $\tau_D = 3\tau_2 = 3.3$ ps[43], which does not match any of the relaxation times commonly observed in the decomposition of the dielectric function of water[25]. Our result, however, is in good agreement with an earlier experiment, probing the optical second harmonic generation efficiency of water after THz excitation, which is observed to relax with a time constant of 1.03 ps in the respective experimental data (in contrast to the value of 13 ps given in the text of this work)[44]. OKE studies on liquid water are dominated by a stretched

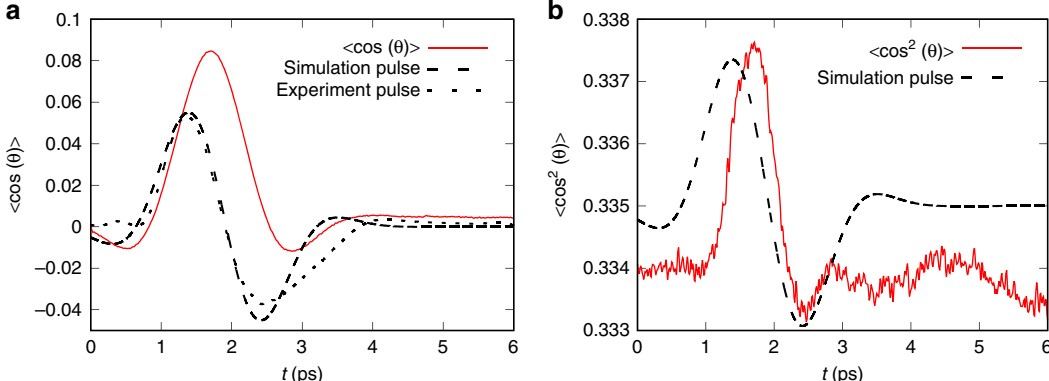

**Fig. 2** MD simulation of the THz Kerr effect in water. Value of (**a**) $\langle \cos(\theta) \rangle(t)$ and **b** $\langle \cos^2(\theta) \rangle(t)$ during the non-equilibrium MD simulation of water. For comparison, the THz pump pulse profiles employed in the experiment and the simulations are shown as black dashed and dotted lines, respectively. $\theta$ is the angle of the water dipoles with respect to the field polarization $\mathbf{u}_z$. $\langle \dots \rangle$ represents the ensemble average over all molecules in the system and all trajectories employed

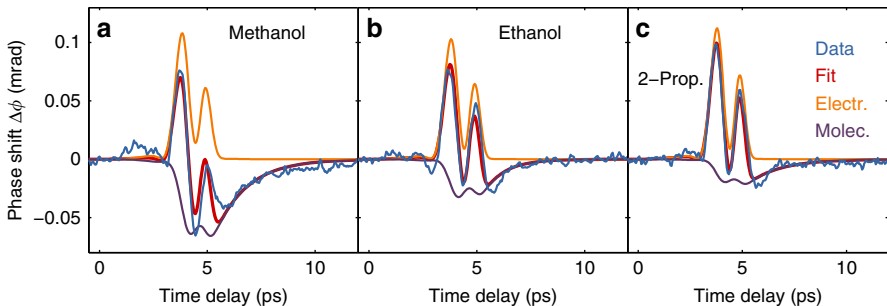

**Fig. 3** The THz-induced Kerr effect in alcohols: **a** methanol, **b** ethanol, and **c** 2-propanol. The color code corresponds to the one in Fig. 1. While all alcohols also reveal a negative polarizability anisotropy, they also show a THz-induced electronic response, which clearly dominates, e.g., in 2-propanol

exponential relaxation, $\exp[-(t/\tau_0)^\beta]$[45,46], with $\tau_0 = 1.00$ ps at ambient conditions. In an earlier study, a bi-exponential relaxation of the OKE was reported with time constants of 0.9 and 2.5 ps[47].

For comparison, we have performed the same experiment on heavy water ($D_2O$), the results of which are depicted in Fig. 1d and equally show a negative $B_m^{(2)}$. The magnitude of $B_m^{(2)}$ for heavy water is 20% smaller, which is explained by calculating the molecular Kerr coefficient[48] in Eq. 5,

$$K^{(m)} = \frac{6nV_m\lambda}{(n^2+2)^2(\epsilon+2)^2} B_m^{(2)}, \qquad (5)$$

with $n$ the refractive index at the probe wavelength, $\epsilon$ the real part of the dielectric function in the THz regime, and $V_m$ the molar volume. Using the numbers given in Table 1, we obtain $K^{(m)} = -1.83 \times 10^5$ cm$^5$/V$^2$ both for regular and heavy water and can conclude based on their equal dipole moments, that their $\Delta\alpha$ is also similar. The only parameter that differs significantly between normal and heavy water is the relaxation time $\tau_2$, which increases by 24% to 1.36 ps in heavy water—consistent with an increase of the molecular mass and inertia, and in good agreement with the 27 or 30% increase in relaxation time reported in literature[49,50].

**THz Kerr effect in aqueous solutions.** Dissolving sodium iodide (NaI) with molarities of 1, 3, 5, and 9.5 in water changes the relaxation time $\tau_2$ by less than 10%. This is fully consistent with earlier observations by THz time-domain spectroscopy, finding that weakly hydrated ions such as Na$^+$ and I$^-$ have little influence on the water reorientation dynamics[51]. The magnitude of the molecular orientation mechanism $B_m$, however, increases

linearly with the concentration of sodium iodide to $B_m = -0.10 \times 10^{-14}$ mV$^{-2}$ at 9.5M. This increase in the Kerr coefficient, reported here, implies an increase of $-\Delta\alpha\mu^2$, i.e., either the dipole moment increases, or the polarizability becomes more anisotropic. Considering that an increase of the dipole moment corresponds to the localization of electronic charge, a decrease in the polarizability in the direction of the dipole moment $\alpha_\parallel$ is to be expected. Since this corresponds to an increase of $-\Delta\alpha$, we expect that both the anisotropy of the polarizability and the dipole moment increase due to the addition of NaI.

**THz Kerr effect in alcohols.** The Kerr effect of alcohols is shown in Fig. 3a–c, including a decomposition into electronic and molecular contributions. While the molecular component $B_m^{(2)}$ turns out negative for all alcohols investigated, this effect is almost compensated by a positive electronic response. In a very recent study of the TKE in methanol, using pulses with ~1 THz center frequency for excitation, Kampfrath et al. observed a positive Kerr effect[52], suggesting that the molecular response mechanism depends critically on the driving frequency. Due to the strongly non-zero dipole moments of the alcohols, the negative sign of $B_m^{(2)}$ implies that this is a molecular orientation mechanism as well and therefore $\Delta\alpha < 0$—in case of methanol consistent with earlier calculations using density functional theory (DFT) and a polarizable force field[9]. In ethanol, dielectric saturation, i.e., a field-induced decrease of the dielectric constant was also observed experimentally[16]. The magnitude of the TKE as represented by the molecular Kerr coefficient $K^{(m)}$ decreases monotonically with increasing size of the molecules, namely $K^{(m)}$

$= -(7.2, 5.0, \text{ and } 3.4) \times 10^5 \text{ cm}^5/\text{V}^2$ for methanol, ethanol, and 2-propanol, respectively.

## Discussion

We have shown that single-cycle electromagnetic pulses in the THz regime orient the dipole moments of liquid water along their electric field. Given the resulting negative sign of the birefringence $\Delta n = n_\parallel - n_\perp < 0$, we provide experimental evidence that the polarizability of regular and heavy water molecules in the liquid state is lower parallel to their dipole moment than perpendicular, i.e., $\alpha_\parallel < \alpha_\perp$. Sodium iodide enhances the THz-induced birefringence without changing the relaxation time of the orientation mechanism. While our Langevin-based model describes the dynamics of the alcohols and non-polar molecules fully, it fails to reproduce the measured THz-induced dynamics of water, motivating further studies of the influence of underlying parameters. The ultrafast orientation of water reported for the first time has the potential to provide further insight into the transient structure of water corresponding to the mode excited by the THz pulse. Our results will assist the modeling of water molecules and provide a benchmark for ab-initio simulations of the electronic structure.

## Methods

**Experimental setup.** The experimental setup is schematically depicted in Supplementary Fig. 1. It utilizes optical pulses from a Ti:Sapphire chirped-pulse laser amplifier with fundamental wavelength of 800 nm, 150-fs pulse duration, and 7-mJ pulse energy to generate THz pulses by optical rectification in LiNbO₃. The optical pulse-fronts are tilted to fulfill the phase-matching condition[53]. The THz pulses generated from this source are de-magnified using two off-axis parabolic mirrors with 4″ and 3″ focal lengths. In the image plane, the pulses are characterized by electro-optic (EO) sampling using a 50-μm-thick <110>-cut GaP and a 200-μm-thick <110>-cut ZnTe crystal. The electric field waveform consists of a single cycle with peak electric field strength of 510 kV/cm and 0.25-THz center frequency (see Supplementary Fig. 5). The THz beam diameter in the focus is 1 mm—an order of magnitude larger than the optical probe spot. For the measurement of liquid samples, Spectrosil® (synthetic fused silica) cuvettes are used with 1-cm-diameter aperture and two 1.2-mm-thick windows enclosing a 0.2-mm-thick sheet of liquid. Only for CS₂, due to its low THz absorption coefficient, a cuvette of 2-mm inner thickness was used. To measure the optical birefringence, the polarization of the probe beam is tilted by 45° with respect to the THz electric field polarization. A Kerr effect time trace is recorded by scanning the delay between the pump (500-Hz repetition rate) and the probe (1 kHz), reading the pump-induced modulation detected by the balanced photodiodes using a lock-in amplifier. The raw data so obtained are depicted in Supplementary Figs. 3 and 4.

**Terahertz time-domain spectroscopy.** Due to the group-velocity mismatch between the optical and THz pulses, it is important to consider the dielectric function in the THz regime for each liquid. Therefore, we measure the complex dielectric functions of all liquids in the same geometry, using a dedicated commercial setup for THz time-domain spectroscopy (TDS). The resulting dielectric functions are depicted in Supplementary Figs. 7 and 8. In the process of extracting these data, the influence of the additional interfaces with the windows of the cuvette were removed by evaluating the complex transfer function of this geometry (see Supplementary Note 3, where the procedure is described in detail).

**Sample source.** All liquids except neat water were obtained commercially with >99.9% purity for methanol and 2-propanol, >99.8% purity for ethanol, benzene and D₂O, and >99% for CS₂. Ultra-pure water was obtained from a lab-based purification system, specified to <0.1 μS/cm at 20 °C. During all measurements, the liquid was held at a constant temperature of 296 ± 1 K.

**Data availability.** Raw experimental data are available from the corresponding author upon request.

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

## Acknowledgements

This work has been supported by the excellence cluster 'The Hamburg Centre for Ultrafast Imaging - Structure, Dynamics and Control of Matter at the Atomic Scale' of the Deutsche Forschungsgemeinschaft (by grant EXC 1074), the priority program QUTIF (SPP1840 SOLSTICE) of the Deutsche Forschungsgemeinschaft and European XFEL. P.Z. and C.B. thank Prof. Thomas Kühne for valuable discussions and P.Z. gratefully acknowledges discussions with Dr. Tobias Kampfrath and Dr. Janne Savolainen. L.S. gratefully acknowledges support through an ONCPR fellowship from China and the Helmholtz Association.

## Author contributions

TKE measurements were carried out by P.Z., L.S. and H.H. on the setup built by L.S., X. W., O.D.M., F.X.K. and P.Z. THz-TDS experiments were performed by P.Z., A. and L. S. MD simulations were done by P.K.M., R.W. and R.S., and the Langevin model was calculated by J.R. and M.T. Experimental data were analyzed by P.Z., who wrote the manuscript with input from all authors. The project was conceived by C.B., P.Z., F.X.K. and R.S.

## Additional information

**Competing interests:** The authors declare no competing interests.

