## [Peer Review File · Nature Communications]

Reviewer #1 (Remarks to the Author):

The manuscript “Molecular polarizability anisotropy of liquid water revealed by terahertz – induced transient orientation” reports on very interesting and elaborate experiments probing the birefringence in liquid water, aqueous NaI solutions and alcohols induced by a strong THz pulse. I believe that the experimental data are extremely interesting and in particular the different sign of the observed phase shift for water and alcohols as compared to benzene and CS₂ deserves – in my opinion – publication in Nature Communications itself. However, there seem to be some inconsistencies in the assignment of the observed dynamics, which also cast some doubts on the extraction of the molecular polarizability anisotropy. Thus, I recommend revision of analysis and interpretation and a more detailed discussion of the observed dynamics before publication in Nature Communications. Details follow below:

My main concern is related to the interaction of the THz pulse with the samples. For instance, on page 3 the authors write: “The relaxation time constants of these mechanisms are explained by the hopping of molecules to unoccupied sites in the tetrahedral structure (8.31 ps), the orientation relaxation time of a single molecule (1.0 ps) and the vibrational relaxation of hydrogen bonds (0.10 ps) 36. With its relaxation time of 1.0 ps, the orientation relaxation mechanism has the highest imaginary susceptibility at 0.2THz 36, and therefore is expected to couple efficiently to our electric field pulses.” Part of this statement above is as far as I can say wrong: Inspection of figure 5 of Reference 36 shows that the mode assigned to single molecule rotation (1ps) has NOT the highest imaginary susceptibility, at 0.2THz, but the contribution of the 8ps relaxation to the imaginary permittivity is 3x larger.

Thus the question that remains open to me is: how exactly does the THz field perturb water? This is in my opinion very important for judging if the subsequent analysis based on single molecular diffusion and molecular polarizability.

- The decay time of the birefringence for water is reported to ~ 1ps, which seems to be somewhat consistent with dominance of the 1ps dielectric mode described above (though there might be a difference between 1st and 2nd order Legendre polynomial correlation functions). This mode has been assigned to single molecule rotation in Reference 36. The molecular origins of the spectral contributions at 0.2 THz are however very controversially discussed (see e.g. Phys. Rev. Lett. 107, 117601, 2011; PHYSICAL REVIEW E 96, 062607 (2017)). For instance, the cage and caged dynamics reported in Phys. Rev. Lett. 107, 117601, 2011 do not necessarily mean that water’s dipole is aligned. Thus, the used single molecular properties in the analysis authors’ analysis may not be appropriate. Similar ideas are put forward in Ref 42. Also Kampfrath et al (Ref 29) seem to interpret their results on the THz induced birefringence in part along those lines.

- Alternatively, the dominant interaction between the field and water is dominated by the 8ps mode, which one might expect based on the higher susceptibility (see above and ref 36). I presume that the observed decay of the birefringence with 1ps is too fast to be consistent with this scenario and I would expect for this case that the decays are slower (cf. In ref 29 DMSO is found to decay bi-exponentially, with 2ps assigned to structural relaxation). The authors may also want to compare and discuss their results in light of the early work of Hochstrasser (Chemical Physics Letters 309 1999 221–228), who detected THz induced second harmonic generation, and found a much slower decay of the signals. As Hochstrasser's experiments use a similar perturbation scheme of water, I am surprised that the decay is much slower for the second harmonic signal. Maybe the authors can elaborate on the experimental differences?

Given the ambiguity in the THz interaction with water, I am concerned about the subsequent analysis to extract the polarizability: It is not clear to me that dipole-field interaction and resulting alignment of water dominates the signal as other perturbations of water may well contribute. I am also not fully convinced that the single molecular diffusion formalism that is used in Part IV of the SI is appropriate for highly collective THz modes. This doubt could possibly be ignored if the agreement between model and experiments was perfect, however, the agreement is not very good.

I think all these potential pitfalls should be discussed in more detail in a revised version and possibly the conclusion might be that the experiments cannot be used to extract the polarizability anisotropy. Nevertheless, I find the experiments extremely interesting and they have the potential to elucidate water dynamics at sub-ps timescales. Thus I recommend publication of the reported experimental work. I also anticipate that this is very difficult to a complex system like water and alcohols and my concerns and the related questions cannot be fully resolved. Possibly, additional experiments on liquids less complex than water but more complex than CS₂ could help shedding light on the observed dynamics (e.g. dipolar liquids that are less correlated like acetonitrile or benzonitrile)

Some additional remarks:

- the authors also use NaI solutions and in the authors conclude that NaI does not affect water significantly. One needs to be a bit careful, as at THz frequencies also cage fluctuations of the ions can contribute (see figure 4b of The Journal of Chemical Physics 141, 214502 (2014)). Hence at such high concentrations the ions will likely contribute and the discussion in terms of unchanged water polarizability at high concentrations is not fully justified.

-One major challenge, as mentioned by the authors, is the low signal intensity for water and alcohols. Did the authors try to use different cuvette materials, which would help the reader to judge on the reliability of the cuvette subtraction?

- I find the discussion of the DC field experiments in the introduction (3rd paragraph of the introductions) rather confusing and misleading as these experiments study very different electric field frequencies. This should either be clearly mentioned or the discussion should be omitted.

Reviewer #2 (Remarks to the Author):

The authors present an original approach to study an important and very fundamental property of the ubiquitous, yet every surprising, liquid water, as well as other polar liquids. The manuscript is clearly written. Before I can recommend publication, I would like to ask for some clarifications:

1. There is large body of literature on dielectric relaxation phenomena of liquid water and the agreement on the three mentioned timescales and their interpretation is not as clear as presented here, see e.g. PRL 107, 117601 (2011). Furthermore, it is the ~ 8 ps one that is most generally assigned to reorientation phenomena, as confirmed for instance by ultrafast polarization-resolved infrared pump-probe studies, where timescales of ~ 2.5 ps are typically found (corresponding to the ~ 8 ps dielectric relaxation timescale, since a different order of the correlation function is probed). Can the authors assess to what extent their results are independent of exact timescales and interpretations of the modes? Also, have they tried different THz frequencies, e.g. one closer to the sub-picosecond mode or (probably more complicated) one closer to the reorientation mode at 20 GHz?

2. Do the authors have an explanation for the deviation between data and model for the liquid H₂O and D₂O cases? Could it be related to the timescales and their assignment?

3. The authors claim that the NaI solutions do not have a very different dielectric relaxation time scale, which is mostly correct. However, the Na⁺ cations do lead to a significant fraction of water dipoles that are 'immobilized', and so-called anisotropic reorientation (see works of R. Buchner and H.J. Bakker). What will be the implication of this for the results of this work? Is it consistent? Related to this: it would be very interesting to try different ion combinations to study for example effects related to the Hofmeister series.

Reviewer #1		
	The manuscript “Molecular polarizability anisotropy of liquid water revealed by terahertz – induced transient orientation” reports on very interesting and elaborate experiments probing the birefringence in liquid water, aqueous NaI solutions and alcohols induced by a strong THz pulse. I believe that the experimental data are extremely interesting and in particular the different sign of the observed phase shift for water and alcohols as compared to benzene and CS₂ deserves – in my opinion – publication in Nature Communications itself. However, there seem to be some inconsistencies in the assignment of the observed dynamics, which also cast some doubts on the extraction of the molecular polarizability anisotropy. Thus, I recommend revision of analysis and interpretation and a more detailed discussion of the observed dynamics before publication in Nature Communications.	We appreciate the excitement the referee shares about our results and are grateful for clearly pointing out his concerns, which we have addressed in our detailed response below and in a revised manuscript as follows.
1	My main concern is related to the interaction of the THz pulse with the samples. For instance, on page 3 the authors write: “The relaxation time constants of these mechanisms are explained by the hopping of molecules to unoccupied sites in the tetrahedral structure (8.31 ps), the orientation relaxation time of a single molecule (1.0 ps) and the vibrational relaxation of hydrogen bonds (0.10 ps) 36. With its relaxation time of 1.0 ps, the orientation relaxation mechanism has the highest imaginary susceptibility at 0.2THz 36, and therefore is expected to couple efficiently to our electric field pulses.” Part of this statement above is as far as I can say wrong: Inspection of figure 5 of Reference 36 shows that the mode assigned to single molecule rotation (1ps) has NOT the highest imaginary susceptibility, at 0.2THz, but the contribution of the 8ps relaxation to the imaginary permittivity is 3x larger. Thus the question that remains open to me is: how exactly does the THz field perturb water? This is in my opinion very important for judging if the subsequent analysis based on single molecular diffusion and molecular polarizability.	We have rewritten the discussion of our results to distinguish more clearly the conclusions drawn from the various results. Our MD simulations, performed with the same water model as in [Phys. Rev. Lett. 107, 117601, 2011], explicitly considers the THz electric field and shows that the main perturbation mechanism is an orientation of the permanent dipole moments along the electric field. Since correlation effects beyond single molecular diffusion are described by MD simulations, we therefore know that the THz-induced perturbation of water is dominated by orientation. This is consistent with our experimental observation of a negative molecular Kerr effect, which can be explained only by the orientation mechanism, albeit under the assumption of isotropic diffusion. To mitigate uncertainties caused by the subtraction of the signal from the cuvette we have performed additional temperature-dependent measurements, now included in Figure 1 of the manuscript, which unequivocally (since background-free) resolve the negative sign of the molecular Kerr effect. We are therefore convinced that we can report an orientation mechanism of liquid water. Regarding the decomposition of the susceptibility into relaxation mechanisms: We agree with the reviewer that the mechanism with 8 ps Debye relaxation time should dominate the response at 0.25 THz and have included a more complete summary of previous literature on this topic. We note that the old (and misleading) statement quoted by the reviewer was intended to say that the orientation mechanism peaks at 0.2 THz – not that it dominates at 0.2 THz and has been rewritten.
2	- The decay time of the birefringence for water is reported to ~ 1ps, which seems to be somewhat consistent with dominance of the 1ps dielectric mode described above (though there might be a difference between 1st and 2nd order Legendre polynomial correlation functions). This mode has been assigned to single molecule rotation in Reference 36. The molecular origins of the spectral contributions at 0.2 THz are however very controversially discussed (see e.g. Phys. Rev. Lett. 107, 117601, 2011; PHYSICAL REVIEW E 96, 062607 (2017)). For instance, the cage and caged dynamics reported in Phys. Rev. Lett. 107, 117601, 2011 do not necessarily mean that water’s dipole is aligned. Thus, the used single molecular properties in the analysis authors’ analysis may not be appropriate. Similar ideas are put forward in Ref 42. Also Kampfrath et al (Ref 29) seem to interpret their results on the THz induced birefringence in part along those lines.	Indeed, the Langevin model in eqs. 2&3 is related to the Debye relaxation times with a factor of three, so that our experimental observation of a 1.1 ps relaxation time corresponds to a Debye relaxation time of 3.3 ps. We have rewritten this part of the discussion and find that this Debye relaxation time is not reported in any of the recent, consistent identifications of these modes in new references 31-34, including also [Phys. Rev. Lett. 107, 117601, 2011], new reference 35. However, the relaxation time of 1 ps is consistent with several earlier optical Kerr effect experiments now summarized in more detail in the manuscript. Please note that we now also discuss earlier observations of caged, correlated dynamics, but find no indication for their contribution on the THz-induced Kerr effect.
3	- Alternatively, the dominant interaction between the field and water is dominated by the 8ps mode, which one might expect based on the higher susceptibility (see above and ref 36). I presume that the observed decay of the birefringence with 1ps is too fast to be consistent this scenario and I would expect for this case that the decays are slower (cf. In ref 29 DMSO is found to decay bi-exponentially, with 2ps assigned to structural relaxation). The authors may also want to compare and discuss their results in light of the early work of by Hochstrasser (Chemical Physics Letters 309 1999 221–228), who detected	The work by Hochstrasser is highly relevant in this context and we discuss it in the revised version of this manuscript. Most interestingly, their data show a decay time constant of 1.03 ps, in perfect agreement with our work, even though they quote a time constant of 13 ps, which does not correspond to the data they show, see a digitized fit to their experimental data below:

THz induced second harmonic generation, and found a much slower decay of the signals. As Hochstrasser's experiments use a similar perturbation scheme of water, I am surprised that the decay is much slower for the second harmonic signal. Maybe the authors can elaborate on the experimental differences?	 (Black lines are guides to the eye with the lower one located at 1/e of the vertical position of the one above.)
Given the ambiguity in the THz interaction with water, I am concerned about the subsequent analysis to extract the polarizability: It is not clear to me that dipole-field interaction and resulting alignment of water dominates the signal as other perturbations of water may well contribute. I am also not fully convinced that the single molecular diffusion formalism that is used in Part IV of the SI is appropriate for highly collective THz modes. This doubts could possibly be ignored if the agreement between model and experiments was perfect, however, the agreement is not very good.	We agree that the deviations between model and experiment in the old version of the manuscript are significant. We have therefore performed additional measurements of the TKE at different temperatures of the sample. Since the Kerr effect of the cuvette and the electronic Kerr effect of water are temperature-independent, the remaining temperature-dependent contribution originates from the molecular orientation mechanism alone. We now present these data in a revised version of the manuscript and find that they agree well with our model. It is worth mentioning that also the changes of the TKE signal for heavy water as compared to normal water are consistent with the validity of a single molecular picture: The relaxation time changes according to the increased rotational inertia expected for the heavier molecule.
I think all these potential pitfalls should be discussed in more detail in a revised version and possibly the conclusion might be that the experiments cannot be used to extract the polarizability anisotropy. Nevertheless, I find the experiments extremely interesting and they have the potential to elucidate water dynamics at sub-ps timescales. Thus I recommend publication of the reported experimental work. I also anticipate that this is very difficult to a complex system like water and alcohols and my concerns and the related questions cannot be fully resolved. Possibly, additional experiments on liquids less complex than water but more complex than CS₂ could help shedding light on the observed dynamics (e.g. dipolar liquids that are less correlated like acetonitrile or benzonitrile)	While we hope that our responses above have clarified most of these concerns, we would like to point out that we demonstrate that our model accurately describes the TKE of more complex liquids like the alcohols as seen in Fig. 3 of the manuscript.
- the authors also use NaI solutions and in the authors conclude that NaI does not affects water significantly. One needs to be a bit careful, as at THz frequencies also cage fluctuations of the ions can contribute (see figure 4b of The Journal of Chemical Physics 141, 214502 (2014)). Hence at such high concentrations the ions will likely contribute and the discussion in terms of unchanged water polarizability at high concentrations is not fully justified.	We agree that the assumption of an unchanged charge distribution along the dipole moment is not sufficiently supported by previous reports at the high concentrations employed in the present work. We have therefore adapted this conclusion in the manuscript.
-One major challenge, as mentioned by the authors, is the low signal intensity for water and alcohols. Did the authors try to use different cuvette materials, which would help the reader to judge on the reliability of the cuvette subtraction?	We have not tested cuvettes of different material but have performed additional temperature-dependent measurements, allowing us to perform background-free measurements. Since Fig. S9 shows that the Kerr effect from the cuvette is fully temperature-independent, the remaining signal must originate from water alone.
- I find the discussion of the DC field experiments in the introduction (3rd paragraph of the introductions) rather confusing and misleading as these experiments study very different electric field frequencies. This should either be clearly mentioned or the discussion should be omitted.	We agree that the water response in the DC regime is expected to be quite different from the THz regime and have therefore removed the introductory paragraph and now only mention in the discussion that using DC fields, also a reduction of the refractive index in the direction of the applied field is observed.

Reviewer #2		
	The authors present an original approach to study an important and very fundamental property of the ubiquitous, yet every surprising, liquid water, as well as other polar liquids. The manuscript is clearly written. Before I can recommend publication, I would like to ask for some clarifications:	We are grateful to the reviewer for sharing our excitement about the importance of the properties under investigations and for carefully studying our work as well as providing clear remarks on how to improve the manuscript.
1	There is large body of literature on dielectric relaxation phenomena of liquid water and the agreement on the three	Most importantly, our MD simulations, performed with the same water model as used in PRL 107, 117601 (2011) and

	mentioned timescales and their interpretation is not as clear as presented here, see e.g. PRL 107, 117601 (2011). Furthermore, it is the ~ 8 ps one that is most generally assigned to reorientation phenomena, as confirmed for instance by ultrafast polarization-resolved infrared pump-probe studies, where timescales of ~ 2.5 ps are typically found (corresponding to the ~ 8 ps dielectric relaxation timescale, since a different order of the correlation function is probed). Can the authors assess to what extent their results are independent of exact timescales and interpretations of the modes? Also, have they tried different THz frequencies, e.g. one closer to the sub-picosecond mode or (probably more complicated) one closer to the reorientation mode at 20 GHz?	explicitly considering the THz electric field, reveal an orientation of the water molecules to be the dominating response. Also, the negative molecular Kerr effect, now supported by additional background-free measurements, can only be caused by an orientation of the water dipole moments. We have rewritten the respective sections to clarify this chain of arguments. Therefore, our results are independent of the interpretations of the relaxation modes observed in dielectric spectroscopy. As suggested also by reviewer #1, we have revised the discussion of relaxation times. Indeed, the relaxation times reported in our work are related to the Debye relaxation times by a factor of 3, so that the relaxation time of water (1.1 ps) reported here would imply a Debye relaxation time of 3.3 ps. Earlier pump-probe experiments such as the work by Cook et al. (new ref. 54) and Winkler et al. (new ref. 57) observe relaxation times of 1 ps (and 2.5 ps in case of Winkler et al.), which corresponds to a Debye relaxation time of 3 ps (and 7.5 ps). The same experiment as the one by Winkler was later refined by a stretched exponential also with a time constant of 1 ps (new ref. 55). We have revised the manuscript accordingly. We have not repeated the experiment at different THz-pump frequencies since the tunability of these sources is very limited and always leads to a significant reduction of the field strength, which we cannot afford given the limited signal/noise levels. Nevertheless, such experiments are in preparation to be performed at tunable THz sources (where commonly user time must be requested).
2	Do the authors have an explanation for the deviation between data and model for the liquid H₂O and D₂O cases? Could it be related to the timescales and their assignment?	Our temperature-dependent TKE measurements resolve this question, since they are sensitive only to the molecular contribution to the Kerr effect in water. The resulting data are in good agreement with the molecular response obtained from the model. Therefore, this deviation must originate from an electronic Kerr effect either in the cuvette or in water. This is now discussed in detail in the manuscript.
3	The authors claim that the NaI solutions do not have a very different dielectric relaxation time scale, which is mostly correct. However, the Na⁺ cations do lead to a significant fraction of water dipoles that are 'immobilized', and so-called anisotropic reorientation (see works of R. Buchner and H.J. Bakker). What will be the implication of this for the results of this work? Is it consistent? Related to this: it would be very interesting to try different ion combinations to study for example effects related to the Hofmeister series.	We agree with the reviewer that a detailed study of the TKE along the Hofmeister series is a promising experiment and we consider this a future study. Due to the unknown impact of Na⁺ cations and possible further effects caused by the cage dynamics, we have decided to revise the discussion of the impact of NaI in the manuscript. Indeed, the weakly hydrated ions Na⁺ and I⁻ were reported by Tielrooij et al. [Journal of Physical Chemistry B, 115, 43, 12638 (2011)] to have little impact on the reorientation dynamics of water. We have added this consistent observation to the manuscript and thank the reviewer for bringing this to our attention.

Reviewer #1 (Remarks to the Author):

In the revised version of the manuscript by Zalden et al addressed nearly all of my previous concerns. There is only one part where I disagree with the authors, this could however be readily solved if the authors tone down a few statements in the manuscript. Thus, I recommend publication in Nature Communications after the following minor revisions:

- I only disagree with the causality in the authors comparison of the MD simulations and the experiments: While the simulations indeed show that the THz field aligns dipoles, the simulations do not evidence that this alignment dominates the Kerr effect traces. Thus, I believe that statements like

“simulations confirm that the THz-induced perturbation is dominated by an orientation”

“dominating contribution of the orientation-term”

have to be toned down.

An alternative explanation might be that the THz pulse also excites (populates) the H-bond bending vibration at ~ 2 THz. Thus, the H-bond angle would be distorted and the conclusions would be somewhat different. I am not saying this is the case, but the MD simulations do also not exclude this.

- I would appreciate if the authors make the reader aware of the discrepancy between the 10ps decay in the text and the 1ps decay of the data in the Hochstrasser paper (I was very surprised by that).

- Meanwhile Kampfrath et al have published a study on alcohols (10.1021/acs.jpcllett.7b03281). I suggest adding a short note and compare the present findings to their findings when the results of the alcohols are discussed.

Reviewer #2 (Remarks to the Author):

The authors have adequately answered my questions and concerns. I recommend publication as is.

Reviewer #1 (Remarks to the Author):

In the revised version of the manuscript by Zalden et al addressed nearly all of my previous concerns. There is only one part where I disagree with the authors, this could however be readily solved if the authors tone down a few statements in the manuscript. Thus, I recommend publication in Nature Communications after the following minor revisions:

- I only disagree with the causality in the authors comparison of the MD simulations and the experiments: While the simulations indeed show that the THz field aligns dipoles, the simulations do not evidence that this alignment dominates the Kerr effect traces. Thus, I believe that statements like "simulations confirm that the THz-induced perturbation is dominated by an orientation" "dominating contribution of the orientation-term" have to be toned down. An alternative explanation might be that the THz pulse also excites (populates) the H-bond bending vibration at ~2THz. Thus, the H-bond angle would be distorted and the conclusions would be somewhat different. I am not saying this is the case, but the MD simulations do also not exclude this.	We have toned down the statements mentioned by the reviewer, which are the only occasions where we state that the orientation mechanism of the permanent dipole moments dominates the THz-induced Kerr effect. The new statements are: "simulations confirm that the THz-pulses induce an orientation of the water molecules" and "existence of an underlying orientation mechanism", as highlighted in the revised manuscript file.
- I would appreciate if the authors make the reader aware of the discrepancy between the 10ps decay in the text and the 1ps decay of the data in the Hochstrasser paper (I was very surprised by that).	We have now included this remark in the text of the manuscript.
- Meanwhile Kampftrath et al have published a study on alcohols (10.1021/acs.jpcclett.7b03281). I suggest adding a short note and compare the present findings to their findings when the results of the alcohols are discussed.	We have included a reference to this work on two occasions, where it fits into the context of the present work.

Reviewer #2 (Remarks to the Author):

The authors have adequately answered my questions and concerns. I recommend publication as is.